# Effects on Child Development and Parent–Child Interaction of the FACAM Intervention: A Randomized Controlled Study of an Interdisciplinary Intervention to Support Women in Vulnerable Positions through Pregnancy and Early Motherhood

**DOI:** 10.3390/ijerph21050587

**Published:** 2024-05-02

**Authors:** Maiken Pontoppidan, Lene Nygaard, Jonas Cuzulan Hirani, Mette Thorsager, Mette Friis-Hansen, Deborah Davis, Ellen Aagaard Nohr

**Affiliations:** 1VIVE—The Danish Centre for Social Science Research, Herluf Trolles Gade 11, 1052 Copenhagen, Denmark; jjh@vive.dk (J.C.H.); meje@vive.dk (M.T.); mfh@vive.dk (M.F.-H.); 2Research Unit for Gynecology and Obstetrics, Department of Clinical Research, University of Southern Denmark, 5230 Odense, Denmark; lene.nygaard1@rsyd.dk (L.N.); eanohr@health.sdu.dk (E.A.N.); 3Department of Gynaecology and Obstetrics, Odense University Hospital, 5230 Odense, Denmark; 4Faculty of Health, University of Canberra and ACT Health, Bruce, ACT 2617, Australia; deborah.davis@canberra.edu.au

**Keywords:** pregnant, mother, family, mental health, infant, multidisciplinary, early intervention, disadvantaged populations, parenting, randomized controlled trial

## Abstract

Health inequality can have a profound impact on a child’s life. Maternal mental health challenges can hinder bonding, leading to impaired functioning and poorer child outcomes. To provide extra support for vulnerable pregnant women, the FACAM intervention offers the services of a health nurse or family therapist from pregnancy until the child starts school. This study examined the effects of FACAM intervention on pregnant women in vulnerable positions and their children until the child turned two years old. We randomly assigned 331 pregnant women to either FACAM intervention or care as usual and assessed them at baseline and when the infant was 3–6, 12–13.5, and 24 months old. The primary outcome was maternal sensitivity measured by Coding Interactive Behavior (CIB). Secondary outcomes included the parent–child relationship, child social–emotional development, child developmental progress, parent–child interaction, and child development. Our findings indicate that care-as-usual children were significantly more involved than FACAM children when the child was 4–6 months old (b = −0.25, [−0.42; −0.08] d = −0.42). However, we suspect this result is due to a biased dropout. We did not find any significant differences in any other outcomes. Therefore, the study suggests that the FACAM intervention is not superior to care as usual regarding child development and parent–child interaction outcomes.

## 1. Introduction

Fetal life and infancy are profoundly significant for a child’s development and are characterized by rapid and foundational changes across multiple domains. The infant’s brain undergoes a remarkable transformation, with synaptic connections forming and the brain’s intricate architecture taking shape, all influenced by the child’s interactions with its environment [1,2,3]. Ideally, an environment that supports infant development should be characterized by nurturing, consistent, and protective interactions with adults [2]. Conversely, children exposed to neglect, abuse, or other forms of toxic stress in their early years may experience long-term consequences, including health issues, attachment difficulties, developmental challenges, mental health disorders, and poorer educational outcomes when compared to their peers [3,4,5,6,7,8,9].

Bonding with the infant is crucial for parents as it establishes a foundation for healthy emotional, social, and cognitive development in both the caregiver and the child. The development of the caregiver–infant relationship starts during pregnancy and develops further after the infant is born. The attachment forms the basis of the child’s internal working model, shaping their expectations and perceptions of relationships throughout their lives [2,10,11]. Developing a secure attachment is important as securely attached infants demonstrate better emotional regulation and coping mechanisms, effectively handling stress and adversity [12].

The transition into parenthood is marked by rapid psychological, physiological and social transformations, which can present challenges for parents. Parents who are at higher risk of adversity due to significant financial, social, or emotional concerns, including mental health conditions such as depression, anxiety, borderline personality disorder, post-traumatic stress disorder (PTSD), or schizophrenia, may have a harder time bonding with the fetus and newborn child. Additionally, parents who have experienced childhood trauma, including neglect and abuse, may bear lasting physical and mental health repercussions [13] challenging their new role as parents and may have a higher risk of inadvertently neglecting their own children [14,15,16].

The experience of the birth is also important for the parent–infant bonding. Having a delivery different than a spontaneous vaginal delivery can cause negative birth experiences in both parents and hinder the bonding [17]. Women experiencing childbirth-related PTSD may also find it difficult to bond with the infant [18].

Given the importance of the early years on a child’s development, it is imperative to nurture and stimulate a supportive family environment that serves as a foundation for the child’s future wellbeing. Systematic reviews and meta-analyses of early parenting interventions find positive results on child emotional adjustment and behavior, parenting skills, parent mental health, parental sensitivity, and parent–child relationships [19,20,21,22,23,24,25]. Moreover, studies show that early interventions aimed at disadvantaged families are better economic investments than interventions later in life [26,27]. Families experiencing challenges across diverse domains, including mental health, physical wellbeing, parenting, and social issues, require interdisciplinary interventions initiated during pregnancy [28,29]. However, parenting interventions offered during pregnancy are typically aimed at more specific [30,31] health concerns like obesity [32], diabetes [33], smoking cessation [34,35], or the importance of breastfeeding [36].

This trial examines the effect of the parenting intervention the FAmily Clinic and Municipality intervention (FACAM) when offered to pregnant women in vulnerable positions [37]. We hypothesize that FACAM mothers will be more sensitive to their child’s signals compared to CAU mothers. We also hypothesize that mothers with more risk factors present at baseline will benefit more from the intervention than mothers with less risk factors. This paper focuses on the effects on the mother–child relationship (primary outcome) and the development and wellbeing of the child at ages 3–6, 12–13.5, and 24 months, which is the most intensive phase of the intervention.

## 2. Materials and Methods

The study was a prospective, superiority, parallel, 1:1 randomized controlled trial where 331 pregnant women were randomized to receive either the intervention (FACAM) or care as usual (CAU). The study was approved by the Health Research Committee in the Southern Denmark Region (journal no. 18/48509) and received ethical approval from the internal review board at VIVE—the Danish Center for Social Science Research. The Committee on Health Research Ethics in the Region of Southern Denmark assessed the protocol and found no need for further study approval (Case no. S-20182000-110). The trial conforms to the Consolidated Standards of Reporting Trials (CONSORT) statement.

### 2.1. Participants

Participants were pregnant women in care groups 3 or 4 according to the Danish health authorities’ recommendations for antenatal care [38,39]. Care groups 3 and 4 include women with complex or severe social, psychological, and psychiatric problems, previous or current harmful use of legal or illegal addictive drugs and alcohol, or concern for the parent’s ability to take care of the child.

#### Inclusion and Exclusion Criteria

We included pregnant women at least 15 years old who fulfilled the following criteria: living in Odense municipality, antenatal care group 3 or 4 according to the Danish health authorities’ recommendations, and enrolled in the family clinic at Odense University Hospital. Women were excluded if they were pregnant with twins, were unable to fill out questionnaires in Danish or English, experienced a life-threatening illness in parent or child, or if the family participated in the FACAM project with an older child. Women were withdrawn from the study if the child was placed in out-of-home care.

### 2.2. Intervention and Comparison

#### 2.2.1. Care as Usual

In the control group, participants received standard care. In Denmark, antenatal care, baby health checkups, and social services are free of charge and provided according to the woman’s needs. For women facing severe mental health issues, significant social issues, or harmful substance abuse, additional checkups by a specialist team are provided, typically at a family clinic [38,40]. This extended care is implemented due to the elevated risk of preterm birth and other pregnancy-related complications [41,42,43]. For pregnant women in care groups 3 and 4, the routine prenatal care package typically involves four to seven midwife consultations, three general practitioner appointments, and two ultrasound scans [40].

Additionally, women with high-risk pregnancies receive tailored care to address their specific needs, including consultations with a social worker, a medical doctor, and a therapist at the family clinic. Uncomplicated births are supervised by midwives in hospital settings. Following hospital discharge, the family’s municipality of residence is notified about the birth, and the family is then offered standard home visits.

Health visitors employed by the municipality provide home visits and adhere to the guidelines issued by the Danish National Board of Health [44]. Municipalities can provide additional services to families needing further assistance, such as extra home visits or interventions to support parenting. All Danish health visitors are registered nurses with specialized training encompassing 1.5 years, focusing on maternal, child, and family health. In the first year of a child’s life, all mothers are offered a post-birth checkup and three well-child checkups with a general practitioner.

#### 2.2.2. The FACAM Intervention

Intervention families received the FACAM intervention in addition to their standard care. Based on a need for an early, interdisciplinary approach to support pregnant women in vulnerable situations and their families, the FACAM intervention was developed in 2017–2018 by a collaborative project team from Odense Municipality and the Family Clinic at Odense University Hospital. The FACAM intervention assigns a dedicated support person (referred to as the FACAM person) to each pregnant woman from pregnancy until the child reaches school age. This flexible intervention is tailored to each woman’s specific needs, drawing on principles of mentalization and strategies aimed at reducing health disparities [45,46].

Additionally, it leverages the well-researched benefits of continuity of care, particularly within midwifery practices [47]. The role of the FACAM person is to guide and refer families to appropriate support resources, whether from hospitals, childcare services, or relevant volunteer organizations. The goal is to offer pregnant women and mothers easy access to practical assistance and support from a trusted professional, enabling them to dedicate more attention to nurturing their children.

The intervention is based on guidelines outlining the FACAM person’s responsibilities and contact frequency (Danish guidelines can be obtained from the corresponding author). These tasks encompass accompanying participants to healthcare and social care visits during and after pregnancy and consultations with professionals like midwives, general practitioners, social workers, or job consultants. The FACAM person can also conduct extra home visits or make phone calls based on the family’s specific needs. Contact with families can involve addressing practical matters such as vaccination reminders, guidance on daycare registration, and discussions regarding financial matters, family dynamics, health, contraception, child-rearing, and mother-child attachment, among others.

During pregnancy and the child’s first year, the FACAM person could provide up to 47 h of support to the family. All FACAM participants were also offered an attachment-based course during pregnancy and the first months of the child’s life. Low-concern families were offered eight two-hour sessions of the Circle of Security parenting group program (COS-P) when the child was around 2 months old, whereas medium or high-concern families could receive up to 50 h of individual sessions focused on attachment, including an attachment interview. After the child’s first year and until they start school at age 6, the FACAM person could offer up to 10 h of support each year.

The FACAM persons were health visitors or family therapists. Family therapists are specialized mental health professionals working with families and hold backgrounds such as pedagogues, social workers, or psychologists. FACAM persons completed a four-day training focusing on mentalization and took part in various one-day courses covering topics like mental health and third-sector organizations. Additionally, they received supervision from a clinical psychologist. Health visitors were also trained in the Alarm Distress Baby Scale (ADBB) method. In families where professionals had a low level of concern, a health visitor served as the FACAM person, also taking on the family’s regular health visitor role. However, for families where the level of concern was higher, a family therapist functioned as the FACAM person. In such cases, regular home visits were conducted by a health visitor (who may not have been trained in FACAM).

Participants in both groups could receive additional care during the trial. If they chose to relocate from Odense Municipality, the FACAM intervention was discontinued.

### 2.3. Procedures and Randomization

We recruited 331 pregnant women to the study—163 to FACAM and 168 to CAU. The flow chart is presented in Figure 1.

During the first visit to the family clinic, the midwife introduced the study to the pregnant woman, provided recruitment flyers, and obtained written consent. At this initial appointment, the midwife categorized mothers into four levels of concern: (1) High concern (if there was a report to Child Protective Services about the family), (2) Medium concern (if it was likely that a report to child protective services would occur during pregnancy), (3) Low concern (when a family might benefit from an attachment-based course with few other concerns), and (4) Minimal to no concern about the family.

Following consent, a research team member administered the baseline questionnaire. Upon questionnaire completion, the participant was randomized to FACAM or CAU (1:1) using the randomization tool in REDCap [48]. An independent data analyst generated the randomization sequence before recruitment began. Participants were stratified into two groups based on the midwife’s concern assessment (Levels 1 and 2 as high concern, Levels 3 and 4 as low concern). Once randomized, the participant and the municipality’s project coordinator were informed of the assigned group. FACAM persons were matched with intervention families, prioritizing those with available capacity in the same geographical area. Once allocated, the intervention commenced. The consent form and related documents can be obtained from the authors upon request.

#### Blinding

Given the additional support in the intervention group, participants and care providers could not be blinded to group allocation. However, outcome assessors, coders, and data analysts remained blinded to allocation status.

**Trial registration:** The study was registered on 6 September 2018, at Clinicaltrials.gov NCT03659721. https://clinicaltrials.gov/ct2/show/NCT03659721 (accessed on 17 April 2024).

### 2.4. Data Collection

Data were collected via web surveys at five time points. T0: baseline, T1: baseline part 2 around gestational week 25, T2 when the child was 3 months old, T3 when the child was 12 months old, and T4 when the child was 24 months old. Furthermore, a 6 min video of interaction between mothers and their child was recorded at the health visitor routine visit or in the home when the child was 4–6 months old. At T3, when the child was 13.5 months old, a research assistant assessed child development and recorded a second video of mother and child interaction at a municipal location or during a home visit. Finally, at T4, a childcare teacher assessed the child’s socio-emotional development.

Survey data were collected through a secure online survey database, REDCap [48], hosted at OPEN, Odense University Hospital, Region of Southern Denmark. REDCap logged data entry and verification. Participants received an email with a direct questionnaire link through e-Boks, a secure digital mailbox system used by Danish citizens. Reminders were sent every 3 days by email. The research team provided phone support for mothers needing assistance with the questionnaire. Mothers received a 200 DKK (~25 EUR) electronic gift card at each data collection. Data were transferred to secure servers hosted by The Agency for Governmental IT Services (Statens IT), adhering to ISO27001 standards for information security. Access to the complete dataset was limited to the trial statisticians, principal investigator, co-PI, and senior investigator.

### 2.5. Outcomes

Table 1 shows the timing of the outcomes included in this paper.

A detailed description of all measures can be found in the protocol paper [38].

#### 2.5.1. Baseline Measures

Socio-demographic measures included the mother’s age, education, occupation, ethnicity, number of children, household status, housing situation, household economy, substance abuse, breastfeeding expectations, childbirth weight, child gestation at birth, breastfeeding, and child health. In addition to these measures, we also included the following baseline assessments to assess initial levels and consider them as potential moderators or confounders in the effect analyses: Prenatal Parental Reflective Functioning Questionnaire (P-PRFQ) [49], The Warwick-Edinburgh Mental Wellbeing Scale (WEMWBS) [50,51], Hospital Anxiety and Depression Scale (HADS) [52,53], Experiences in Close Relationship Scale-Short Form (ECR-S) [54], Adverse Childhood Experience (ACE), and PTSD-8 [55].

#### 2.5.2. Outcomes

This section outlines the outcomes included in this paper. The primary outcome is maternal sensitivity, which is assessed at a child age of 12 months using the Coding Interactive Behavior (CIB) instrument [56]. The CIB, a global rating system for social interactions, includes 22 parent codes, 16 child codes, and 5 dyadic codes, rated on a scale of 1 to 5. These codes can be aggregated into composites: sensitivity, intrusiveness, limit setting, involvement, withdrawal, compliance, dyadic reciprocity, and dyadic negative states. Mother–infant interactions were recorded during a 6 min free play session. The CIB system has been validated as an assessment measure in various studies of mother–child interactions, demonstrating stability, predictive validity, and adequate psychometric properties [56,57,58,59]. Cronbach’s alpha at T3 is 0.90 for sensitivity, 0.69 for intrusiveness, 0.86 for limit setting, 0.83 for involvement, 0.54 for withdrawal, 0.95 for dyadic reciprocity, and 0. 80 for dyadic negative states. The coding was conducted by the first author (MP), an expert coder, and two trained coders (coder 1 and coder 2). The inter-coder agreement was assessed using a randomly selected 10% sample subset. At T2, inter-coder agreement was 93% between the two coders (N = 19), and at T3, inter-coder agreement was 91% between the expert coder and coder 1, and 95% between the expert coder and coder 2 (N = 22).

#### 2.5.3. Secondary Outcomes

The remaining composites of the CIB measured the parent–child relationship: intrusiveness, limit setting, involvement, withdrawal, reciprocity, and negative states.

Ages and Stages Questionnaire-Social Emotional 2 (ASQ:SE-2) [60] measures child social-emotional development consisting of seven subscales: self-regulation, compliance, social communication, adaptive functioning, autonomy, affect, and interaction with people. Total score ranges from 0–150 (3 months, 15 items) to 0–260 (12 months, 26 items). Cronbach’s alpha is 0.62–0.79 for the total score and ranges from 0.05 to 0.58 for the subscales. Due to the low reliability of the subscales, we only use the total score in the analyses. A low score indicates better development.

Ages and Stages Questionnaire 3 (ASQ:3) [61] is a 30-item measure of child developmental progress consisting of five subscales: communication, gross motor, fine motor, problem-solving, and personal-social. Cronbach’s alpha is 0.56 for communication, 0.56 for gross motor, 0.76 for fine motor, 0.71 for problem-solving, and 0.60 for personal–social. The total score ranges from 0 to 300, and a low score indicates better development.

Activities with the child consist of 4 items constructed for this study to measure parent–child interaction through activities such as singing and reading. The total score ranges from 4 to 24; a high score indicates more interaction. Cronbach’s alpha is 0.59 for the total score.

The Mother and Baby Interaction Scale (MABISC) [62] is a 10-item measure of the mother–infant relationship. Cronbach’s alpha is 0.74. The total score ranges from 0 to 40; a high score indicates a better relationship.

Bayley Scales of Infant and Toddler Development 3rd Edition -Screening Test (BSID) is a test to assess child development [63]. The BSID consists of three primary subtests: cognitive, language, and motor scales. Raw scores for each subscale are converted into scaled scores (range 1–19, M = 10, SD = 3). A composite score (M = 100, SD = 15) can be derived from the scaled score for cognitive development and the sum of the two language scaled scores. The test was administered at T3 by trained psychology students supervised by an experienced and trained psychologist.

The Social–Emotional Assessment/Evaluation Measure (SEAM) [64] is a measure of child social–emotional development consisting of 35 items covering two indexes: 1) empathy and 2) self-regulation and positive self-image. The score range is 0–66 for empathy (Cronbach’s alpha 0.92), and 0–39 for self-regulation and positive self-image (Cronbach’s alpha 0.78), and a high score indicates higher empathy and better self-regulation and positive self-image

### 2.6. Fidelity

We recorded whether the assigned FACAM person was a health visitor or a family therapist for each FACAM participant. After each visit, the FACAM person completed a brief questionnaire specifying the type of support provided to the family. We also documented participation in the attachment sessions.

### 2.7. Sample Size Justification

The power calculation for the primary outcome relied on a meta-analysis of interventions targeting parenting sensitivity [65]; the overall average effect size was 0.44 (standardized mean difference, SMD), but for randomized trials, the average effect size was 0.33. With normally distributed outcomes and a two-sided alpha of 0.05, our sample of 331 participants (163 FACAM and 168 CAU) yields 85% power to detect an effect size of 0.33. However, given that the primary outcome is based on video observations where the dropout rates are relatively high, the power to detect effect sizes of 0.33 is reduced to 63% and to 86% for effect sizes of 0.44 when the sample size is reduced to 194 participants (103 FACAM and 91 CAU).

### 2.8. Data Analysis

All outcomes were tested using linear regression with robust standard errors since we were interested in the mean difference and allowed for heteroscedasticity in the error term. The treatment effect was estimated using a binary indicator of treatment. Variables with indications (*p* < 0.05) of differences between intervention and CAU groups at baseline were used as control variables. For outcomes based on parental questionnaires, missing data were handled using multiple imputations using all available baseline data. For outcomes based on video interactions (the primary outcome of maternal sensitivity), the Bayley test, and teacher questionnaires, we could not use multiple imputations because non-response rates were too large for imputation to be valid. Multiple imputations are valid when missing data are either missing at random (MAR) or missing completely at random (MCAR). To informally test for MCAR, we checked if missingness was predictable using all available baseline characteristics in a logit model. Out of 42 included baseline characteristics only three variables were significant at the 5% level. The analysis shows that highly educated mothers, mothers with fewer worries about their job situation and mothers who are more depressed are less likely to be missing. This indicates that the data are missing at random although the assumption is inherently untestable. We applied multiple imputations as our main specification for questionnaire data and assessed the robustness of this decision.

The primary analysis was based on the intention-to-treat (ITT) principle, aiming to include all participants in the arm they were initially allocated to, irrespective of the treatment received. Sensitivity analyses were performed to investigate the potential impact of missing data, mainly using a complete case analysis and an Instrumental Variable (IV) approach to account for non-participation in the intervention. Two-sided tests with 0.05 significance levels were applied throughout.

We conducted subgroup analyses to explore potential differences between the following participant subsets: education (high school or less versus more than high school), concern about the family (level 1/2 or level 3/4), initial trauma level (ACE < 3 or ACE ≥ 3) and attendance (dose). We used interaction models, including the subgroup characteristic and the interaction between the treatment and subgroup indicators, for the baseline regression. The interaction term coefficient shows if the treatment effect of the intervention differs across the different subgroups. Due to data limitations, we did not perform the following subgroup analyses (specified in the protocol): primiparous or multiparous, provider (health visitor or family therapist), adult attachment style (ECR-S), the initial level of reflective function (lowest 50% versus highest 50%), the initial level of depression or anxiety (clinical or not-clinical level).

We performed two robustness checks to test the sensitivity of the main specification: data without imputation, and an instrumental variable approach to deal with non-participation in the intervention.

## 3. Results

Of the 562 invited pregnant women, 332 (57%) consented to participate in the trial and filled out the baseline questionnaire. After completion, one participant retracted her consent and wished to have all data deleted, leaving the baseline sample that was randomized to 331 pregnant women (163 FACAM and 168 CAU).

### 3.1. Participant Characteristics

Table 2 presents descriptive statistics for the study population according to allocation group. We observe only one significant difference at baseline: more women in the control group expected their first child (66% versus 54%). The mean age of the participants was 30 years, and 82% cohabited with their partner. For almost half of the participants, high school (12 years of school) or less was the highest education achieved. In total, 36% were employed, and 19% were students. The remaining participants were on sick leave (17%) or without employment or education (28%). Participants had experienced a mean of 2.5 traumatic events in their childhood, and 20% had indications of Post Traumatic Stress Disorder (PTSD). Participants in the group with higher levels of concern (N = 91) were at higher risk on several characteristics (e.g., younger, shorter education, lower rates of employment, higher levels of smoking, childhood trauma, and PTSD symptoms) than the group with lower concern (see Appendix A).

### 3.2. Participation

The FACAM intervention consisted of two parts: visits by the FACAM person and attachment sessions. FACAM participants received a mean of 9.3 FACAM visits (median 7) until the child was 12 months old. Due to various implementation issues (including the COVID-19 pandemic), more than half of the participants (91 or 55%) received no attachment sessions. The total mean number of sessions (FACAM visits and attachment sessions) when the child was 12 months old was 15. Families in the high-concern group received more visits (23) than participants in the low-concern group (12).

### 3.3. Attrition

At baseline, 331 respondents responded to the questionnaire. This was reduced to 284 (questionnaire at child age 3 months), 248 (questionnaire at child age 12 months), 189 (video at child age 4–6 months), 194 (video and Bayley at child age 13.5 months), and 164 (teacher assessment at child age 24 months) (see Figure 1). The dropout rates for the questionnaire data (mother and teacher) are relatively similar across FACAM and CAU groups. However, for video and Bayley data, dropout rates are higher in the CAU group (close to 50%) than in the FACAM group (around 35%). We also find that families in the medium or high concern group are likelier to drop out than families in the low concern group. At baseline, the two groups are well-balanced across background characteristics. However, due to selection in the dropout and the higher dropout in the CAU group, we do observe imbalances between the groups in later follow-ups. We control for these imbalances in the regressions.

### 3.4. Intervention Effects

Table 3 shows the means and standard deviations for the outcomes included in this paper.

Table 4 shows regression output comparing FACAM and CAU mothers. We do not find any significant difference between the two groups for the primary outcome (maternal sensitivity) at any time. For secondary outcomes, we find that CAU children are significantly more involved in the relationship than FACAM children when the child is 4–6 months old (b = −0.25, [−0.42; −0.08] d = −0.42). For the remaining outcomes, we do not find any significant differences between the two groups at any of the time points.

### 3.5. Differential Effects

Due to the relatively large dropout rate, we could only examine the differential effects of three outcomes: level of education, concern, and trauma (Appendix A). Only a few of the analyses show significant differences in the interaction analyses, and the results do not systematically favor one group or the other.

### 3.6. Sensitivity Analysis

Sensitivity analyses included regression without imputation for questionnaire data and instrument variable estimation (Appendix A). The sensitivity analyses’ results confirmed the primary regression analysis results.

## 4. Discussion

This paper investigates the effects of the interdisciplinary FACAM intervention provided to pregnant women in vulnerable positions on outcomes related to child development and the mother–child relationship. The findings indicate no significant effects of the FACAM intervention, except for one outcome where CAU children demonstrate significantly higher involvement in the relationship than FACAM children aged 4–6 months. We attribute this result to potential bias dropout rates in video data. The results of the other CIB constructs favor the CAU group, though significance is not reached. Video data dropout is notably higher for the CAU group than the FACAM group, particularly in families with high concerns. For this subgroup, we only have video data from 30% of the control group compared to 51% for the FACAM group when the children are 4–6 months old. This discrepancy is less pronounced at 13.5 months, with video data from 59% of CAU families in the high-concern group compared to 51% in the FACAM group. Given that families in the high-concern group face challenges such as lower educational levels and more trauma experience, it is concerning if dropout rates significantly differ between the two groups. Consequently, the indications of negative effects of the FACAM intervention on the mother–child relationship are likely a result of biased dropout and may not accurately reflect actual effects.

Participants in this study are often underrepresented in clinical trials, partly due to difficulties in comprehending the uncertainties related to allocation and intervention [66]. Pregnancy serves as a window of opportunity for intervention, given the heightened motivation of pregnant woman to make behavior changes for the wellbeing of their baby [67]. Despite this, many participants chose not to participate, perceiving the FACAM intervention as overly extensive and unnecessary for their specific needs. Some preferred the standard offer, avoiding involvement in a project meant for “specially selected” individuals. Others declined participation, expressing discomfort with feeling “pathologized” and simply desiring an average pregnancy experience. It is also possible that some of the most vulnerable mothers chose not to participate as they may fear potential repercussions, such as their child being removed from their care if they disclose worries or challenges [68].

After recruitment to this study concluded, the FACAM intervention was revised and implemented to target the subgroup subject to the highest concerns. Removing the need for families to provide consent for randomization resulted in a significantly larger proportion opting to participate in the revised FACAM intervention. This underscores the difficulty that families at a higher risk of disadvantage may face in understanding and accepting randomization and other trial logic.

We encountered a significant dropout in both the intervention and control groups, which is not unexpected given the complex problems and challenging life circumstances prevalent in this target group. Individuals facing such challenges often exhibit reluctance to engage with social services and may lack the energy to accept additional support. Systematic reviews examining dropout rates in the treatment of Borderline Personality Disorder (BPD) [69] and Post Traumatic Stress Disorder (PTSD) [70] indicate mean dropout rates of 16% for PTSD and 28% for DBT treatment. They also found that higher dropout rates are linked to longer treatment duration, randomization, and an outpatient setting [70] —all characteristics present in the FACAM study. Reasons for dropout included lack of motivation and dissatisfaction with treatment, and most dropouts occurred in the first half of treatment [70]. To optimize recruitment and minimize dropout, we carefully tailored all processes to cater to this pregnant women group. This included training midwives for recruitment, providing incentives, keeping questionnaires short, assisting with questionnaire completion, and maintaining high flexibility with video recordings and Bayley III assessments.

As anticipated, dropout rates were significantly higher for video data and Bayley III assessments than questionnaire assessments. The nature of video recordings and tests, resembling exams, could create apprehension among parents who feared being judged. To address this concern, we developed additional materials and provided comprehensive information about the procedures for video recordings and Bayley III assessments. This included a short informational video featuring the trial’s principal investigator (first author) to familiarize parents with the process and alleviate any anxieties.

It is possible that the lack of results from the intervention is due to the fact that we were not able to recruit as many families with high levels of concern as we had expected. According to municipal data, we believed that the number of pregnant women with a high level of concern (groups 1 and 2) would be similar to those with a low level of concern (groups 3 and 4), with around 160 of each. Unfortunately, despite our efforts to recruit more high-concern pregnant women, we only managed to recruit 27% of them. Moreover, many of those in the control group with high levels of concern dropped out of the study, which compounded the issue.

Another explanation for the lack of differences between the two groups in our study could be that we were conducting research in a welfare society with a high standard of care for pregnant women and families. The comparison, therefore, was not between a treated and a non-treated group, but rather between two different types of intervention. This means that it is not uncommon for studies in Denmark to find no additional effects of the intervention in pregnant women or families with newborns, as evidenced by published trials such as those by Røhder et al. [71], Pontoppidan et al. [72,73], and Brixval et al. [74], as well as several newer, unpublished studies.

Denmark’s robust infrastructure for healthcare, including prenatal and postnatal care, midwifery services, and early childhood interventions, makes it difficult to discern significant differences between the two interventions. Pregnant women and families in Denmark receive extensive and standardized care as part of standard practice, making it challenging to demonstrate additional effects from specific interventions. Additionally, Denmark’s welfare-oriented policies, such as accessible healthcare services, parental leave policies, and social support mechanisms, contribute to a generally supportive environment for pregnant women and families. In light of these broader societal and healthcare contexts, it is crucial to interpret research outcomes in a nuanced way, particularly in welfare societies like Denmark.

## 5. Conclusions

We investigated the effects of the interdisciplinary FACAM intervention provided to pregnant women in vulnerable positions on outcomes related to child development and the mother–child relationship. Our findings indicate no significant effects of the FACAM intervention when the children are 3, 4–6, 12, and 13.5 months old. The control group received usual care, which may have been relatively similar to the FACAM intervention. We therefore conclude that the FACAM intervention does not appear to be superior to the usual care for this group of participants.

## Figures and Tables

**Figure 1 ijerph-21-00587-f001:**
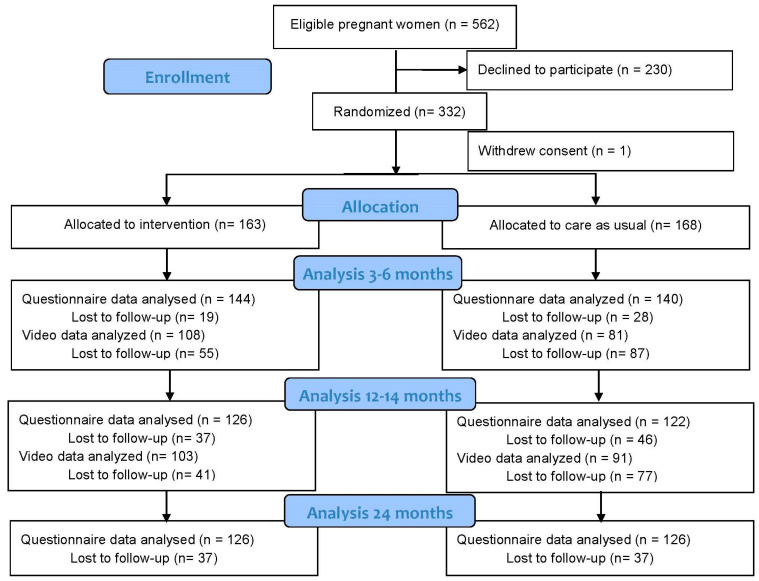
Study flow chart.

**Table 1 ijerph-21-00587-t001:** Timing of the outcomes included in the paper.

		T2	T3	T4
Child measures				
Social–emotional development	ASQ-SE2	√	√	
Child development	ASQ-3	√		
Bayley Scales of Infant Development	BSID-III		√	
Relationship measures				
Learning activities	Singing, reading		√	
Mother and Baby Interaction Scale	MABISC		√	
Coding interactive behavior (video)	CIB	√	√	
Teacher measures				
Social–emotional development	SEAM			√

**Table 2 ijerph-21-00587-t002:** Study population characteristics at baseline.

	CAU N = 168		FAMKO N = 163		Difference	
	Mean	SD	Mean	SD	T-Stat	*p*-Value
Mother age	29.64	(5.65)	29.64	(5.43)	0.01	0.99
Mother health	7.14	(1.76)	7.13	(1.84)	0.07	0.94
Mother life satisfaction	7.79	(1.85)	7.67	(1.88)	0.60	0.55
Mother well-being	23.70	(3.81)	23.81	(4.35)	−0.24	0.81
Ever lonely	2.41	(0.92)	2.58	(0.93)	−1.70	0.09
Access to practical help	4.10	(1.02)	4.00	(1.05)	0.84	0.40
Access to somebody to talk to	4.59	(0.79)	4.48	(0.88)	1.14	0.26
HADS-anxiety	6.81	(3.90)	6.76	(3.81)	0.12	0.91
HADS-depression	4.15	(3.01)	4.64	(3.53)	−1.36	0.18
PTSD total score	13.83	(5.82)	14.01	(6.16)	−0.28	0.78
ECR: Fear of abandonment	19.64	(7.09)	18.30	(6.95)	1.73	0.08
ECR: Fear of intimacy	14.01	(6.88)	13.17	(6.49)	1.13	0.26
PPRFQ opacity of mental states	4.60	(1.22)	4.57	(1.32)	0.21	0.83
PPRFQ reflecting on the fetus-child	5.11	(1.04)	4.98	(1.01)	1.10	0.27
PPRFQ the dynamic nature of mental states	4.56	(1.05)	4.35	(1.14)	1.75	0.08
ACE total score	2.46	(2.26)	2.40	(2.20)	0.21	0.83
Units of alcohol prior to pregnancy	1.62	(2.26)	1.39	(1.97)	0.97	0.33
Units of alcohol during pregnancy	0.01	(0.11)	0.02	(0.19)	−0.74	0.46
Expecting first child	0.68	(0.47)	0.56	(0.50)	2.15	**0.03**
Cohabit with partner	0.81	(0.39)	0.83	(0.38)	−0.44	0.66
Only speak Danish at home	0.83	(0.37)	0.81	(0.39)	0.56	0.58
High school or less	0.41	(0.49)	0.42	(0.49)	−0.12	0.91
Vocational or secondary education	0.19	(0.39)	0.23	(0.42)	−0.82	0.42
College, Bachelor, tertiary or longer education	0.40	(0.49)	0.36	(0.48)	0.80	0.42
Employed	0.36	(0.48)	0.36	(0.48)	−0.09	0.93
Sick leave	0.14	(0.35)	0.20	(0.40)	−1.30	0.20
Unemployment benefit	0.07	(0.26)	0.06	(0.23)	0.60	0.55
Social assistance/unemployment program	0.13	(0.34)	0.20	(0.40)	−1.75	0.08
In education	0.23	(0.42)	0.14	(0.35)	2.13	0.03
Unemployment no benefits	0.03	(0.17)	0.01	(0.08)	1.61	0.11
Smoking regularly	0.11	(0.32)	0.13	(0.34)	−0.60	0.55
Never regularly used drugs like hash, pot, marihuana	0.89	(0.31)	0.86	(0.35)	0.94	0.35
Never regularly used drugs like amphetamine, ecstasy, cocaine, LSD	0.95	(0.21)	0.94	(0.24)	0.55	0.58
Medicine during pregnancy (including non-prescription painkillers)	0.60	(0.49)	0.62	(0.49)	−0.45	0.65
Expect to breastfeed	0.96	(0.19)	0.97	(0.18)	−0.22	0.83

Notes: SD: Standard deviation.

**Table 3 ijerph-21-00587-t003:** Means and standard deviations of the outcomes at T2, T3, and T4.

	T2				T3			
	FACAM N = 144		CAU N = 140		FACAM N = 126		CAU N = 122	
	Mean	SD	Mean	SD	Mean	SD	Mean	SD
Child health	9.16	(1.24)	9.04	(1.26)				
ASQ:3 Communication	47.00	(9.25)	46.08	(10.33)				
ASQ:3 Gross motor	50.89	(9.12)	50.24	(9.13)				
ASQ:3 Fine motor	38.71	(14.96)	39.31	(14.12)				
ASQ:3 Problem solving	47.00	(11.89)	45.97	(12.59)				
ASQ:3 Personal-social	45.18	(10.82)	43.65	(12.48)				
ASQ:SE Totalscore	32.14	(20.07)	33.30	(17.41)	31.72	(17.40)	30.08	(16.37)
ASQ:SE Totalscore incl. worries	36.21	(27.03)	38.26	(24.92)	24.85	(23.14)	24.69	(21.36)
CIB: Sensitivity	3.15	(0.55)	3.02	(0.56)	3.04	(0.45)	2.97	(0.50)
CIB: Intrusiveness	1.71	(0.42)	1.77	(0.44)	1.89	(0.23)	1.93	(0.27)
CIB: Limit-setting	3.38	(0.76)	3.28	(0.77)	3.48	(0.63)	3.34	(0.68)
CIB: Involvement	3.37	(0.54)	3.09	(0.63)	3.46	(0.49)	3.37	(0.48)
CIB: Withdrawal	1.34	(0.48)	1.51	(0.56)	1.32	(0.38)	1.38	(0.43)
CIB: Reciprocity	3.19	(0.66)	3.02	(0.71)	3.36	(0.64)	3.26	(0.72)
CIB: Negative states	1.81	(0.67)	2.02	(0.78)	1.53	(0.69)	1.64	(0.69)
CIB: Compliance					3.54	(0.62)	3.48	(0.74)
Child activities					2.30	(0.76)	2.36	(0.71)
The Mother and Baby Interaction Scale					8.92	(4.22)	9.60	(4.27)
BSID: Cognitive scale					12.16	(2.61)	11.80	(2.30)
BSID: Receptive language scale					9.03	(2.45)	8.65	(2.70)
BSID: Expressive language scale					10.14	(2.29)	10.03	(1.93)
BSID: Language scale					18.99	(3.95)	18.60	(3.96)
BSID: Fine motor scale					11.29	(2.43)	11.85	(2.39)
BSID: Gross motor scale					9.08	(2.61)	9.44	(2.68)
BSID: Motor scale					20.26	(4.27)	21.29	(4.06)
					T4			
					FACAM N = 78		CAU N = 86	
					Mean	SD	Mean	SD
SEAM: Empathy					56.92	(9.66)	57.50	(8.43)
SEAM: Selfregulation and positive self-image					33.89	(4.98)	34.01	(3.92)

Notes: The sample sizes are different for CIB, and BSID outcomes than listed in the heading of the table. CIB at T2: FACAM N = 108, CAU N = 81. CIB at T3: FACAM N = 103, CAU N = 91. BSID: FACAM N = 105, CAU N = 92. SD: standard deviation.

**Table 4 ijerph-21-00587-t004:** Regression output comparing FACAM and CAU outcomes at T2, T3, and T4.

	T2 Child Age 3–6 Months				T3 Child Age 12–14 Months			
	b	CI	P	d	b	CI	P	d
Child health	−0.13	[−0.41, 0.15]	0.36	−0.11				
ASQ:3 Communication	−0.45	[−2.89, 1.98]	0.71	−0.05				
ASQ:3 Gross motor	0.31	[−1.88, 2.50]	0.78	0.03				
ASQ:3 Fine motor	0.93	[−2.61, 4.46]	0.61	0.06				
ASQ:3 Problem solving	−0.98	[−4.04, 2.07]	0.53	−0.08				
ASQ:3 Personal-social	−0.90	[−3.75, 1.96]	0.54	−0.08				
ASQ:SE-2 Total score	1.51	[−2.94, 5.96]	0.50	0.08	−1.29	[−5.50, 2.92]	0.55	−1.29
ASQ:SE-2 Total score incl. worries	2.93	[−3.09, 8.94]	0.34	0.12	0.23	[−4.52, 4.98]	0.92	0.23
Child activities					−0.02	[−0.21, 0.16]	0.80	−0.02
The Mother and Baby Interaction Scale					0.56	[−0.52, 1.65]	0.31	0.56
CIB: Sensitivity	−0.12	[−0.29, 0.05]	0.16	−0.22	−0.06	[−0.20, 0.08]	0.41	−0.12
CIB: Intrusiveness	0.04	[−0.08, 0.16]	0.54	0.09	0.03	[−0.03, 0.10]	0.32	0.14
CIB: Limit-setting	−0.06	[−0.28, 0.16]	0.60	−0.08	−0.14	[−0.33, 0.05]	0.16	−0.21
CIB: Involvement	−0.25	[−0.42, −0.08]	0.00	−0.42	−0.08	[−0.22, 0.06]	0.25	−0.17
CIB: Withdrawal	0.14	[−0.01, 0.29]	0.06	0.27	0.06	[−0.05, 0.17]	0.30	0.14
CIB: Compliance					−0.09	[−0.28, 0.10]	0.35	−0.13
CIB: Reciprocity	−0.15	[−0.36, 0.05]	0.14	−0.22	−0.09	[−0.29, 0.10]	0.35	−0.14
CIB: Negative states	0.18	[−0.02, 0.39]	0.08	0.25	0.11	[−0.08, 0.30]	0.25	0.16
BSID: Cognitive scale					−0.43	[−1.12, 0.26]	0.22	−0.17
BSID: Receptive language scale					−0.34	[−1.07, 0.39]	0.36	−0.13
BSID: expressive language scale					−0.11	[−0.70, 0.49]	0.73	−0.05
BSID: Language scale					−0.37	[−1.51, 0.76]	0.52	−0.09
BSID: Fine motor scale					0.63	[−0.07, 1.32]	0.08	0.26
BSID: Gross motor scale					0.34	[−0.41, 1.08]	0.37	0.13
BSID: Motor scale					1.09	[−0.09, 2.27]	0.07	0.26
					T4 child age 24 months			
SEAM: Empathy					−0.31	[−3.27, 2.66]	0.84	−0.03
SEAM: Selfregulation and positive self-image					−0.05	[−1.37, −1.27]	0.94	−0.01

Notes: b: regression estimate, CI: 95% confidence interval, P: *p*-value, d: Cohen’s d. Observations: T2: Survey N = 324, CIB N = 189; T3: Survey N = 324, CIB N = 194, BSID N = 196; T4: SEAM N = 164.

## Data Availability

The datasets generated and analyzed during the current study are not publicly available to protect participant privacy but are available from the corresponding author upon reasonable request.

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
