# Peer review of "Effects on Child Development and Parent–Child Interaction of the FACAM Intervention: A Randomized Controlled Study of an Interdisciplinary Intervention to Support Women in Vulnerable Positions through Pregnancy and Early Motherhood"

_ijerph, 2024, doi:10.3390/ijerph21050587_

Round 1

Reviewer 1 Report

Comments and Suggestions for Authors

This is a very interesting study with clear challenges.  Some of the mother's comments in the results are particularly revealing about how some of the women feel about support they are offered.  The comments on introducing the FACAM "...in a welfare society with a high standard of care.." do appear to be apt, and it would be most interesting to see the same study in other countries. Revising the strategy to enable greater take-up of the intervention by the women in most need was a logical and efficient use of resource.

Abstract

line 18 please correct the sentence, perhaps: "This study aimed to examine the effects....."

line 25 please give the words "care as usual" and then the acronym in brackets.

Introduction

Line 78 A reference is required to the first publication and explanation of the FACAM intervention.  Perhaps page 4 of your protocol? Pontoppidan, M., Nygaard, L., Thorsager, M., Friis-Hansen, M., Davis, D., & Nohr, E. A. (2022). The FACAM study: protocol for a randomized controlled study of an early interdisciplinary intervention to support women in vulnerable positions through pregnancy and the first 5 years of motherhood. Trials23(1), 73. 

Line 321 Please amend the word conduct to the paste tense conducted: "We conducted subgroups analyses...."

Lines 332 - 334 "We perform two robustness....in the intervention."  It would read more smoothly to be in the past tense.  

Conclusion

Lines 490 - 491, Please add a short summary of the study as a conclusion. 

Comments on the Quality of English Language

Excellent scientific English. Some very simple corrections suggested above.

Author Response

Reviewer 1:

This is a very interesting study with clear challenges.  Some of the mother's comments in the results are particularly revealing about how some of the women feel about support they are offered.  The comments on introducing the FACAM "...in a welfare society with a high standard of care.." do appear to be apt, and it would be most interesting to see the same study in other countries. Revising the strategy to enable greater take-up of the intervention by the women in most need was a logical and efficient use of resource.

Response: Thank you!

Abstract

line 18 please correct the sentence, perhaps: "This study aimed to examine the effects....."

Response: We have corrected the sentence:  “This study examines the effects of FACAM intervention on pregnant women in vulnerable posi-tions and their children until the child turned two years old.”

line 25 please give the words "care as usual" and then the acronym in brackets.

Response: We have revised the text: “We randomly assigned 331 pregnant women to either FACAM intervention or care as usual…”

Introduction

Line 78 A reference is required to the first publication and explanation of the FACAM intervention.  Perhaps page 4 of your protocol? Pontoppidan, M., Nygaard, L., Thorsager, M., Friis-Hansen, M., Davis, D., & Nohr, E. A. (2022). The FACAM study: protocol for a randomized controlled study of an early interdisciplinary intervention to support women in vulnerable positions through pregnancy and the first 5 years of motherhood. Trials, 23(1), 73.

Response: I am sorry but we are not sure about what you are asking for here? We have added the reference to the protocol paper.

Line 321 Please amend the word conduct to the paste tense conducted: "We conducted subgroups analyses...."

Response: we have changed to past tense as suggested.

Lines 332 - 334 "We perform two robustness....in the intervention."  It would read more smoothly to be in the past tense. 

Response: we have changed to past tense as suggested.

Conclusion

Lines 490 - 491, Please add a short summary of the study as a conclusion.

Response: We have added a conclusion: We investigated the effects of the interdisciplinary FACAM intervention provided to pregnant women in vulnerable positions on outcomes related to child development and the mother-child relationship. Our findings indicate no significant effects of the FACAM intervention when the children are 3, 4-6, 12, and 13.5 months old. The control group received usual care, which may have been relatively similar to the FACAM intervention. We therefore conclude that the FACAM intervention does not appear to be superior to care us usual for this group of participants.

Reviewer 2 Report

Comments and Suggestions for Authors

Dear Authors

I read a very important article. You can make some corrections that will surely perfect its form

Adjust the title according to the needs of the magazine (uppercase-lowercase)

In paragraph (lines 51-55) add the PTSD factor as a factor inhibiting the caregiver-infant bond

The type of delivery can also affect the development of the caregiver's bond with the infant. Grow that factor a bit too

Write more specifically the purpose of the study. You can number the individual aims and objectives. 

some words must be bold. It is necessary? 

Table 2 could be omitted and only the statistically significant data reported

about secondary outcomes, could be collapsed into the text

stretches and limitations are missing. 

The conclusion section is missing. However, remove the sentence from the template if you do not write your conclusions.

Good luck! 

Comments on the Quality of English Language

Minor editing of English language required

Author Response

Reviewer 2:

Dear Authors

I read a very important article. You can make some corrections that will surely perfect its form

Response: Thank you!

Adjust the title according to the needs of the magazine (uppercase-lowercase)

Response: We have applied uppercase to the title.

In paragraph (lines 51-55) add the PTSD factor as a factor inhibiting the caregiver-infant bond

Response: We have added PTSD as suggested: “The transition into parenthood is marked by rapid psychological, physiological and so-cial transformations, which can present challenges for parents. Parents’ who are at higher risk of adversity due to significant financial, social, or emotional concerns, in-cluding mental health conditions such as depression, anxiety, borderline personality disorder, post traumatic stress disorder (PTSD), or schizophrenia, may have a harder time bonding with the fetus and newborn child.”

The type of delivery can also affect the development of the caregiver's bond with the infant. Grow that factor a bit too

Response: Thank you for suggesting to add this relevant information. The following text has been added to the introduction: “The experience of the birth is also important for the parent-infant-bonding. Having a delivery different than the spontaneous vaginal delivery can cause negative birth experiences in both parents and hinder the bonding [18]. Women experiencing child-birth-related PTSD may also find it difficult to bond with the infant [19].”

Write more specifically the purpose of the study. You can number the individual aims and objectives.

Response: we have added specific hypotheses: This trial examines the effect of the parenting intervention the FAmily Clinic and Municipality intervention (FACAM) when offered to pregnant women in vulnerable positions. We hypothesize that FACAM mothers will be more sensitive to their child’s signals compared to CAU mothers. We also hypothesize that mothers with more risk factors present at baseline will benefit more from the intervention than mothers with less risk factors. This paper focuses on the effects on the mother-child relationship (primary outcome) and the development and wellbeing of the child at ages 3-6, 12-13.5, and 24 months which is the most intensive phase of the intervention.

some words must be bold. It is necessary?

Response: we used bold to highlight the construct each measure is assessing to make it easy to read through the section. We would prefer to keep it this way but can change it if necessary. 

Table 2 could be omitted and only the statistically significant data reported about secondary outcomes, could be collapsed into the text

Response: We have decided to keep table 2. This is mainly to make sure that all relevant data are available if the study should be included in a systematic review.  

stretches and limitations are missing.

Response: we felt that a large part of the discussion focused on limitations – especially the biased dropout. We therefore did not add a specific limitation paragraph as we would only repeat the issues from the discussion.

The conclusion section is missing. However, remove the sentence from the template if you do not write your conclusions.

Response: We have added a conclusion: We investigated the effects of the interdisciplinary FACAM intervention provided to pregnant women in vulnerable positions on outcomes related to child development and the mother-child relationship. Our findings indicate no significant effects of the FACAM intervention when the children are 3, 4-6, 12, and 13.5 months old. The control group received usual care, which may have been relatively similar to the FACAM intervention. We therefore conclude that the FACAM intervention does not appear to be superior to care us usual for this group of participants.

Good luck!

Response: Thank you!

Reviewer 3 Report

Comments and Suggestions for Authors

Abstract:

-please rewrite this passage "This study aimed examines the effect..."

-"Our findings indicate that CAU children" --> do not use undefined abbreviations in the abstract

- "were significantly more involved than FACAM children when 25
the child was 4-6 months old (b = -0.25, [-0.42;-0.08] d = -0.42)" --> How can you describe 4-5 months, wenn measurements were at 3, 12 an 24months only?? Please correct.

Manuscript:

Line 70: "breastfeeding" is written here in the context of pre-birth, are you sure you don't mean post-partal?

Line 79: "vulnerable positions": do you mean "vulnerable conditions"? 

End of Introduction: Pleas make your hypotheses transparent here: What effects do you expect?

Line 157: " during pregnancy and the first months of the child’s life": Please define how many months.

Figure 1: Both branches of the randomization are named "allocated to intervention"... please correct.

Line 310: Multiple Imputation is only allowed, when the missings are at random (MAR) or completely at random (MCAR). Please run the Missing analysis before using imputed data and re-analyze your data.

321: we conducted

324: we used

332: we performed

Table 4: SEAM outcomes: Please write decimal numbers with a period, not a comma (e.g. 0.84 instead of 0,84)

5. Conclusion

Please writh some concluding words or at least delete the sentence "This section is mandatory"

Author Response

Reviewer 3:

Abstract:

-please rewrite this passage "This study aimed examines the effect..."

Response: We have corrected the sentence: “This study examines the effects of FACAM intervention on pregnant women in vulnerable positions and their children until the child turned two years old.”

-"Our findings indicate that CAU children" --> do not use undefined abbreviations in the abstract

Response: We have revised the text: “We randomly assigned 331 pregnant women to either FACAM intervention or care as usual…”

- "were significantly more involved than FACAM children when 25

the child was 4-6 months old (b = -0.25, [-0.42;-0.08] d = -0.42)" --> How can you describe 4-5 months, wenn measurements were at 3, 12 an 24months only?? Please correct.

Response: Sorry, the text in the abstract was not correct. Questionnaires were collected at 3, 12 and 24 months while video and bayley assessment was when the children were 4-6 and 13.5 months old. The text now reads: “We randomly assigned 331 pregnant women to either FACAM intervention or care as usual and assessed them at baseline and when the infant was ”.

Manuscript:

Line 70: "breastfeeding" is written here in the context of pre-birth, are you sure you don't mean post-partal?

Response: We are trying to describe what interventions offered during pregnancy usually address. We have changed the text to make it clearer: “Families experiencing challenges across diverse domains, including mental health, physical wellbeing, parenting, and social issues, require interdisciplinary interventions initiated during pregnancy [37,38]. However, parenting interventions offered during pregnancy are typically aimed at more specific health concerns like obesity [31], diabetes [32], smoking cessation [35], or the importance of breastfeeding [33].”

Line 79: "vulnerable positions": do you mean "vulnerable conditions"?

Response: We opted for the phrase "pregnant women in vulnerable positions" to highlight the dynamic nature of their circumstances. This phrase emphasizes the situational aspect of vulnerability, acknowledging the various challenges they may face during pregnancy. We also employed this terminology in our protocol paper.

End of Introduction: Pleas make your hypotheses transparent here: What effects do you expect?

Response: we have added specific hypotheses to the text: This trial examines the effect of the parenting intervention the FAmily Clinic and Municipality intervention (FACAM) when offered to pregnant women in vulnerable positions. We hypothesize that FACAM mothers will be more sensitive to their child’s signals compared to CAU mothers. We also hypothesize that mothers with more risk factors present at baseline will benefit more from the intervention than mothers with less risk factors. This paper focuses on the effects on the mother-child relationship (primary outcome) and the development and wellbeing of the child at ages 3-6, 12-13.5, and 24 months which is the most intensive phase of the intervention.

Line 157: " during pregnancy and the first months of the child’s life": Please define how many months.

Response: There was no exact time point at which the attachment-based course should be completed so we cannot define a specific time point.

Figure 1: Both branches of the randomization are named "allocated to intervention"... please correct.

Response: Sorry, we have corrected the mistake.

Line 310: Multiple Imputation is only allowed, when the missings are at random (MAR) or completely at random (MCAR). Please run the Missing analysis before using imputed data and re-analyze your data.

Response: We have added information as follows: “Multiple imputations are valid when missing data are either missing at random (MAR) or missing completely at random (MCAR). To informally test for MCAR we checked if missingness was predictable using all available baseline characteristics in a logit model. Out of 42 included baseline characteristics only three variables were significant at the 5 %-level. The analysis show that highly educated mothers, mothers with fewer worries about their job situation and mothers that are more depressed are less likely to be missing. This indicates that the data is missing at random although the assumption is inherently untestable.”

321: we conducted

Response: we changed to past tense

324: we used

Response: we changed to past tense

332: we performed

Response: we changed to past tense

Table 4: SEAM outcomes: Please write decimal numbers with a period, not a comma (e.g. 0.84 instead of 0,84)

Response: We corrected the mistakes.

  1. Conclusion

Please writh some concluding words or at least delete the sentence "This section is mandatory"

Response: We have added a conclusion: We investigated the effects of the interdisciplinary FACAM intervention provided to pregnant women in vulnerable positions on outcomes related to child development and the mother-child relationship. Our findings indicate no significant effects of the FACAM intervention when the children are 3, 4-6, 12, and 13.5 months old. The control group received usual care, which may have been relatively similar to the FACAM intervention. We therefore conclude that the FACAM intervention does not appear to be superior to care us usual for this group of participants.

Round 2

Reviewer 3 Report

Comments and Suggestions for Authors

In your new text about the missing analysis:

" The analysis show" --> The analysis shows. Please correct.

Thanks for addressing all the issues and improving the paper.

Author Response

The text is corrected.

Thank you